# Effect of Interfacial Ionic Layers on the Food-Grade O/W Emulsion Physical Stability and Astaxanthin Retention during Spray-Drying

**DOI:** 10.3390/foods10020312

**Published:** 2021-02-03

**Authors:** Eduardo Morales, César Burgos-Díaz, Rommy N. Zúñiga, Johanna Jorkowski, Marcela Quilaqueo, Mónica Rubilar

**Affiliations:** 1Doctorate in Engineering Sciences with Specialization in Bioprocesses, Universidad de La Frontera, Avenida Francisco Salazar, Temuco 01145, Chile; e.morales04@ufromail.cl; 2Agriaquaculture Nutritional Genomic Center, CGNA, Temuco, Chile; cesar.burgos@cgna.cl; 3Department of Biotechnology, Universidad Tecnológica Metropolitana, Las Palmeras, Ñuñoa 3360, Chile; rommy.zuniga@utem.cl; 4Master in Food Technology, Technische Universität Berlin, 10623 Berlin, Germany; jo.jorkowski@web.de; 5Department of Chemical Engineering, Scientific and Technological Bioresource Nucleus, BIOREN, Universidad de La Frontera, Avenida Francisco Salazar, Temuco 01145, Chile; marcela.quilaqueo@ufrontera.cl

**Keywords:** multilayer emulsion, physical stability, spray-drying, astaxanthin retention

## Abstract

The utilization of astaxanthin in food processing is considered to be narrow because of its substandard solubility in aqueous matrices and the instability of chemical compounds during the processing of food and the instability of chemical compounds during the processing of food. The investigation sought to evaluate multilayer emulsions stabilized by ionic interfacial layers of lupin protein isolate (LPI), ι-carrageenan (CA), and chitosan (CHI) on the physical stability of the emulsion as well as the retention of astaxanthin during the spray drying process. Primary emulsion (Pr-E) was prepared by adding LPI on oil droplet surfaces containing astaxanthin. The homogenization pressure and cycles to obtain the Pr-E were investigated. The secondary emulsion (Se-E) and tertiary emulsion (Te-E) were elaborated by mixing CA/Pr-E and CHI/Se-E, respectively. Emulsion stability was assessed under different environmental stresses (pH and NaCl). Astaxanthin retention of emulsions was determined immediately after finishing the spray-drying process. The results showed that Pr-E was stabilized with 1.0% (*w/v*) of LPI at 50 MPa and three cycles. Se-E and Te-E were obtained with CA/Pr-E and Se-E/CHI of 70/30 and 50/50% (*w/w*), respectively. The Se-E was the most stable compared to the Pr-E and Te-E when subjected to different pHs; nevertheless, once the NaCl concentration rose, no variations in the ζ-potential of all emulsions studied or destabilization were observed. The Se-E and Te-E derived provided higher astaxanthin retention (>95%) during the spray-drying process compared to Pr-E (around 88%). The results indicated that these astaxanthin multilayer emulsions show considerable potential as a functional ingredient in food products.

## 1. Introduction

Oil-in-water (*o*/*w*) tend to be generally used in the food, pharmaceutical, and cosmetics industries for encapsulating different lipophilic bioactive compounds and enhancing the solubility and stability of said compounds [1]. Emulsifiers behave as stabilizers of emulsion systems, frequently using one-layered membranes that are susceptible to physical instability, particularly when exposed to environmental stresses such as pH changes, ionic strength changes, heating, and drying [2,3]. On the other hand, in the food industry, emulsions undergo a process of dehydration through spray drying to extend the oil (and encapsulated components) shelf life in order to avoid degradation and oxidation on storage [4]. 

The spray drying process is a technique that involves atomizing the o/w emulsion (containing an active component) in hot air, thus enabling rapid removal of the water. There is a widespread use of this technology in the industry of food due to the relatively low cost and short drying timeframe in comparison with the freeze-drying method [5]. Since some active compounds may be exposed in microparticles, spray-drying is regarded as a technology of immobilization instead of an actual technology for encapsulating [6]. Nevertheless, the process of dehydration of o/w emulsions through high temperatures can also modify the interfacial qualities and produce an alteration in the droplets of the emulsion causing the oil to leak [7]. Consequently, a technique for enhancing the physical stability of *w/w* emulsions when exposed to environmental stresses, like high temperatures, involves forming multilayer emulsions composed of several interfacial layers of emulsifiers (protein or surfactant) and/or polyelectrolytes surrounding the drops of oil, applying the layer-by-layer technique [3].

Currently, some studies indicate that the application of vegetable proteins and ionic polysaccharides used to produce multi-layer o/w emulsions based on interfacial engineering technology enables the creation of natural supply systems of improved stability [8]. In multilayer emulsions, a primary emulsion (Pr-E) is produced through the homogenization of the oil and water phases using an emulsifier with a positive or negative charge. Afterward, The Pr-E obtained is blended with a solution of a polyelectrolyte of opposing charge in order to form a secondary emulsion (Se-E). Then the Se-E is added to another solution that contains polyelectrolytes of opposite charge to the former one so that a tertiary emulsion (Te-E) is obtained [9]. In this context, several studies have demonstrated that vegetal proteins and ionic polysaccharides could be utilized to form multilayer lipophilic compound encapsulation systems such as soy protein isolate, and various ionic polysaccharides such as carrageenan [10], flaxseed gum [11], pectin [12], and chitosan-alginate [13]. Likewise, lupin protein isolate with chitosan-xanthan gum and chitosan/alginate/pectin can be used to develop multilayer o/w emulsions [8,14].

In this study, lupin protein isolate (LPI) was used to stabilize and formulate the first layer of Pr-E, given its notable emulsifying qualities, which were determined previously [8,15]. Following this, ι-carrageenan (CA) and chitosan (CHI) coating layers were added over the first layer for preparing Se-E and Te-E, respectively.

CA is a linear sulfated anionic polysaccharide extracted from red algae (Rhodophyceae), typically applied in the food industry for gelling and thickening applications [16]. An advantage of CA is the fact that it can be used directly as a wall material to encapsulate bioactive components [17]. At the same time, CHI is a polysaccharide obtained from the alkaline deacetylation of chitin N, which represents an important component of the shells of crustaceans, like crabs and shrimps [18]. Moreover, CHI is widely applied when forming coatings as it is one of the few available food-grade cationic polysaccharides that can interrelate with anionic polysaccharides by electrostatic interaction [19]. These ionic biopolymers were used to produce layered coatings because of their opposite electrostatic charge, and for their biodegradability and non-toxicity [20]. LPI with CA and CHI as interfacial layers has not yet been used to develop multilayer o/w emulsions as an astaxanthin encapsulation system.

Astaxanthin is a xanthophyll carotenoid that is present in several microorganisms and marine wildlife. The natural sources of astaxanthin are microorganisms, phytoplankton, marine species, and seafood for instance shrimp, lobster, asteroidean, algae, fish, crustacean, trout, krill, red sea bream, and salmon [21]. Nowadays, studies on astaxanthin have increased owing to the demand for natural bioactive compounds for promoting human health. The capacity of antioxidation in astaxanthin is ten times higher than in zeaxanthin, lutein, canthaxanthin, and β-carotene, as well as 100 times higher with respect to α-tocopherol. It provides protection to organisms against a broad variety of diseases with a high degree of safety and tolerability. Astaxanthin presents great potential regarding food, nutraceutical, and pharmaceutical applications [22]. Nevertheless, its applications in different food formulations are rather reduced since this carotenoid degrades rapidly during processing [23]. Therefore, a novel approach used in this study was the utilization of the multilayer emulsion stabilized by LPI, CA, and CHI as a template to encapsulate astaxanthin before the process of spray-drying.

Consequently, the study aimed to evaluate the effect of multilayer emulsions stabilized by ionic interfacial layers of LPI, CA, and CHI on the physical stability of the emulsion and the retention of astaxanthin during the spray-drying process. Multilayer emulsions were produced by sequentially adding CA and CHI to surfaces of LPI-coated oil droplets. The results obtained through this study will allow a greater understanding of the role of multilayer emulsions with respect to the protection of astaxanthin for potential applications in the food powder industry.

## 2. Materials and Methods 

### 2.1. Materials

LPI from yellow lupin seeds (*Lupinus luteus*) was provided by Agriaquaculture Nutritional Genomic Center. The ι-carrageenan was bought from Sigma Aldrich (Merck, St. Louis, MO, USA). Chitosan used in the study was obtained from Xi’an Surnature Biological Technology Co. Ltd. (Shaanzi, China). Sunflower oil was bought in a local market (Santiago, Chile) and astaxanthin oleoresin of *Haematococcus pluvialis* was supplied by Atacama Bio Natural Products S.A. (Iquique, Chile). Maltodextrin (Dextrose Equivalent, DE: 16.5–19.5) was bought from Sigma Aldrich (Merck, St. Louis, MO, USA).

### 2.2. Interfacial Tension (IT)

The IT at the o/w interface for different concentrations of LPI (from 0.1 to 2.0% *w/v*) has been determined by an optical tensiometer model 250-F4 (Ramé-Hart Instrument CO, Roxbury, Boston, MA, USA) at ambient temperature. To measure IT, the “pendant drop” (a drop of LPI solution dispersed in oil) method was employed. It entails taking a picture of the dispersion droplet using a high-speed digital camera and examining its measurements. An axisymmetric drop (8 μL) of LPI dispersion was provided and placed at the tip of the needle inside a quartz recipient with 30 mL of oil phase over a period of time to cause the adsorption of the emulsifier at the oil-water interface. The interfacial tension (mN/m) was measured using the DropImage software (version, DropImage Advanced, Roxbury Boston, USA) by adjusting the Laplace equation to the shape of the drop. The measurement of each sample was performed in triplicate.

### 2.3. Elaboration of O/W Multilayer Emulsions with Encapsulated Astaxanthin

The oil phase was prepared with a 10% (*w/w*) astaxanthin and 90% (*w/w*) sunflower oil. The Pr-E was obtained by mixing a 10% (*w/w*) oil phase with 90% (*w/w*) protein emulsifier (from 0.1 to 2.0% *w/v* of LPI) at pH 3. The mixture was homogenized using a high-speed blender (Benchtop homogenizer, Pro Scientific Inc. Oxford, USA) at 10,000 rpm for 5 min, and afterward, it was blended in a high-pressure homogenizer (HPH) (PandaPlus 2000, GEA Niro Soavi, Parma, Italy). The impact of HPH process conditions on the size of particles and the polydispersity index of Pr-E was determined. Different homogenization pressures (10 to 70 MPa) and the number of cycles (1 to 5) were analyzed. 

The multilayer emulsions were elaborated in the following manner: the secondary emulsion (Se-E) was elaborated with a mixture of CA solutions (0.1–0.4% *w/v*, at pH 3) with Pr-E (CA/Pr-E) at different proportions (50/50, 60/40, 70/30, and 80/20% *w/w*). Then, tertiary emulsion (Te-E) was produced with a mixture of solutions of CHI (0.1–0.4% *w/v* at pH 3) and Se-E (CHI/Se-E) at different proportions (50/50, 60/40, 70/30, and 80/20% *w/w*). The homogenization of Se-E and Te-E was carried out at 5000 rpm for 2 min each one, after which two cycles at 15 MPa were carried out using an HPH. After that, the emulsions were homogenized at room temperature for 1 h with a magnetic stirrer.

### 2.4. Mean Size of Particles and Polydispersity Index (PdI) Measurements

The mean size of particles and Polydispersity Index (PdI) of the emulsions were estimated through the dynamic light scattering in a laser diffractometer (Zetasizer NanoZS, Malvern Instruments, Worcestershire, UK). The emulsions were diluted before the examination with milli-Q water at a proportion of 1:100 (*v/v*) in order to prevent multiple dispersion effects. The PdI value refers to a measurement of the particle size distribution. Analyses were conducted in triplicate.

### 2.5. ζ- Potential Measurements

The ζ-potential of the emulsions was examined through an instrument combining dynamic light scattering and electrophoresis (Zetasizer NanoZS, Malvern Instruments, Worcestershire, UK). Therefore, the emulsions were dissolved with milli-Q water before analysis in a proportion of 1:100 (*v/v*), then placed in zeta cells made of plastic (DTS 1061, Malvern, UK) and equilibrated at 25 °C during 60 s. These examinations were carried out three times.

### 2.6. Optical Microscopy of Emulsions

The observation of the recently prepared emulsions was carried out with an optical microscope (Olympus-BX43, Tokyo, Japan) to determine the size of the drops and the state of aggregation of oil drops in each emulsion. Therefore, a drop of the emulsion was put on the microscope slide with a coverslip. Images of the samples were obtained with software to process digital images (Micro Video Instruments Inc., Avon, MA, USA).

### 2.7. Creaming Index (CI) of Emulsions

The stability of the emulsion regarding creaming was analyzed with the Creaming Index (CI) method [24]. For this purpose, 5 mL of each emulsion were poured into glass tubes and sealed to ensure that the moisture did not evaporate. The CI (%) of the emulsions was measured during storage for 7 days at 25 °C. All emulsions were diluted to the same concentration to compare their effects during the storage period. Analyses were performed in triplicate. CI (%) was calculated according to Equation (1):(1)$$\mathrm{CI}\;\left(\%\right)=\left(H_{S}/H_{e}\right)\times 100\%$$
where H_e_ represents the total height of the emulsion (mm) and H_s_ refers to the height of the cream layer (mm), which was visually determined in function of time.

### 2.8. Effect of Environmental Stress on the Stability of Emulsions

The effect of different pH values (3–7) was tested on the properties of the emulsions. The pH was modified with the addition of an HCl or NaOH solution. Also, the emulsions with a difference in NaCl concentration (0–300 mM) were elaborated blending the initial emulsions with different proportions of salt and buffer solution. The emulsions prepared were stored under ambient temperature for 24 h before the analysis of the ζ-potential.

### 2.9. Spray-Drying of Emulsions 

The emulsions were placed in a B-290 mini-spray dryer (Büchi, Flawil, Switzerland). Previous to spray drying, maltodextrin was added to the emulsions to obtain a total solids content of 15% and then these mixtures were stirred for 1 h. The spray dryer was used at an inlet air temperature of 120 and 160 °C, 4 mL/min of feeding rate and 85% of aspiration rate. The microcapsules were stored at −20 °C for later characterization.

### 2.10. Assessment of Astaxanthin Content 

The astaxanthin content of emulsions was assessed after the spray-drying process in accordance with the methodology defined by Burgos-Diaz et al. [4]. In the analysis, 0.5 g of emulsion powder was diluted in 5 mL of distilled water and vortexed for 40 min. Following, 5 mL of this extract was then blended with 20 mL of hexane-2-propane (2:1 *v/v*) and vortexed for 1 min. The organic phase was isolated/detached by centrifugation at 1000× *g* for 5 min. Astaxanthin was assayed in the supernatant by Ultraviolet–visible (UV-Vis) spectrophotometry (Synergy HT, BioTek Instruments Inc., Winooski, VT, USA) at 478 nm with a microplate reader. Pure hexane was utilized as a blank in all determinations. The astaxanthin content was estimated, by applying a calibration curve using Equation (2):(2)$$\mathrm{Astaxanthin}\;(\mathrm{mg}/g\;\mathrm{sample})=\frac{\left(\left(A_{\mathrm{sm}}-0.045\right)/4.085\right)\times V}{W_{\mathrm{sm}}}$$
where A_sm_ stands for absorbance of the sample, W_sm_ stands for the mass of the sample in grams and V represents the dilution volume (mL).

### 2.11. Astaxanthin Retention of Emulsions

The retention of encapsulated astaxanthin was determined immediately after finishing the spray-drying process [25]. The astaxanthin retention was realized in triplicate, with Equation (3):(3)$$\mathrm{Astaxanthin}\;\mathrm{retention}\;\left(\%\right)=C_{2}/C_{1}\times 100\%$$
where C_1_ represents the theoretical content of astaxanthin in the powder emulsions (8.5 mg Astax/g emulsion), and C_2_ indicates the content of astaxanthin after finishing the spray-drying process.

### 2.12. Analysis of Statistics

The results shown represent the average and the standard deviation estimated according to these replicate measurements. An analysis of variance (ANOVA) was conducted using a significance level fixed at 0.05. The Duncan test was conducted If relevant disparities were identified using the ANOVA. The statistical analysis was conducted using InfoStat software.

## 3. Results and Discussion

### 3.1. Interfacial Tension (IT) of LPI

Interfacial properties of proteins are essential in order for o/w emulsions to be formed and stabilized [3]. On this basis, Figure 1 represents the process of adsorption of LPI at the oil-water interface regarding the decrease in Interfacial Tension (IT). 

Figure 1 illustrates that the IT started to decrease after a few seconds once the analysis began, thus initiating the process of LPI adsorption at the oil-water interface. The IT of all samples containing LPI at different concentrations decreased considerably (*p* < 0.05) when the time of protein adsorption at the oil-water interface increased from 0 to 800 s. The function of an emulsifier like proteins is to lower the interfacial tension covering the surface of the new-formed oil droplets with a protective layer and to stabilize the droplets in the continuous aqueous phase [26]. According to Miller [27], the IT begins to decrease sharply once a particular interfacial coating of the protein at the oil-water interface is reached, a stage known as the penetration period, completely saturating the oil droplet in the emulsion. However, the results suggested that IT was not altered notably (*p* > 0.05) when time increased from 800 to 3600 s. These results showed that the period of LPI saturation at the oil-water interface is from 800 s onwards, given that relatively constant interfacial tension values were achieved. During this phase, the protein was located at the oil-water interface, where it could adopt a new structural configuration in which polar groups interact with water, and non-polar groups interact with oil, a process known as interfacial denaturation [27].

On the other hand, the results have shown that different concentrations of LPI can be adsorbed by the oil/water interface, as demonstrated by the reduction in the IT of the same interface when proteins are absent (0% LPI). Between LPI concentrations of 0.1 and 0.5% (*w/v*) the decrease in IT in function of time was not considerably different (*p* > 0.05); however, IT as function of time was substantially reduced (*p* < 0.05) as the LPI concentration rose from 0.1 to 1.0% (*w/v*). This decrease in IT as LPI concentration increases could be associated with faster absorption of proteins on the oil droplet surface [3]. Finally, the IT was not affected significantly (*p* > 0.05) when there was an increase of LPI concentration from 1.0 to 2.0% (*w/v*). Therefore, it was demonstrated that 1.0% of LPI would be sufficient to saturate the oil-water interface of the Pr-E. 

### 3.2. Effect of High-Pressure Homogenization Process Conditions on Pr-E 

The impact of homogenization pressure, the number of cycles on mean particle size, and PdI of oil droplets of the Pr-E were evaluated (Figure 2). 

The emulsion was prepared by adding 10% (*w/w*) of the oil phase with 90% (*w/w*) of the emulsifier solution containing 1% (*w/v*) of LPI. The results revealed that particle size (Figure 2a) and PdI (Figure 2b) of oil droplets were not significantly affected (*p* > 0.05) when homogenization pressure increased from 10 to 30 MPa. However, particle size PdI was reduced considerably (*p* < 0.05) when homogenization pressure rose from 30 to 50 MPa. Moreover, there was no relevant difference observed (*p* > 0.05) when the homogenization pressure enhanced from 50 to 70 MPa. 

Particle size and PdI were reduced substantially (*p* < 0.05) when the number of homogenization cycles increased from one to three cycles for all homogenization pressures. Despite that, the increase in the number of homogenization cycles from three to five did not significantly affect (*p* > 0.05) the particle size and PdI. Therefore, the homogenization process conditions selected to prepare the Pr-E were 50 MPa and three cycles, resulting in a particle size of 0.87 ± 0.07 µm. At the same time, the PdI value of 0.583 ± 0.05 suggests partial polydispersity of oil droplets in the emulsion. Similar behavior was reported in previous studies with homogenization pressures (from 30 to 100 MPa) on the particle size (below 1 μm) in the o/w emulsions prepared with cod protein isolate [28] and lentil protein isolate [29] as emulsifiers. A PdI value around 0.1 represents monodispersity, and values near 1 indicate polydispersity [30]. According to Juttulapa et al. [31], high pressures and homogenization cycles can decrease droplet size to less than 1 μm and enhance the shelf-life of the o/w emulsions by decreasing the creaming rate. Stoke’s law indicates that the cremation rate is proportional to the square of the radius of the droplet, for that reason, when droplet size is reduced, the rate of gravitational separation decreases, and consequently the stability increases [32]. 

Pr-E samples had a ζ-potential of 42.0 ± 1.85 mV, which is possibly caused by the positively charged LPI layer surrounding the oil droplets. Systems with high ζ-potential absolute values (>+25 mV or <−25 mV) show, predominantly, great stability because of strong repulsive electrostatic forces between the droplets [33].

### 3.3. Effect of CA and CHI on the Multilayer Emulsion Stability 

The addition of CA on Pr-E (CA/Pr-E) and CHI on Se-E (CHI/Se-E) needed in different ratios to fully coat the oil droplets of the emulsion was assessed (Figure 3). More specifically, the concentration at which the oil droplets of the emulsion are enveloped by the polyelectrolytes was measured [8].

Figure 3a illustrates the effect of adding CA/Pr-E in different ratios needed to obtain a stable Se-E. The results revealed that the ζ-potential absolute value of the oil droplets in the emulsion showed significant alteration (*p* < 0.05) when the addition of CA on Pr-E increased. The oil droplet ζ-potential value was increased considerably (*p* < 0.05) when the addition of CA/ Pr-E increased from 50/50 to 60/40% (*w/w*). Furthermore, between 60/40 and 80/20% (*w/w*), no considerable difference (*p* > 0.05) were perceived regarding the negative electrical charge, indicating that a constant ζ-potential value was obtained from 60/40% (*w/w*). However, some oil droplet aggregations were visualized (data not shown) in the sample tube of emulsions formed with 60/40% (*w/w*) unlike the emulsions obtained between 70/30 and 80/20% (*w/w*). Presumably, in the 60/40% (*w/w*) ratio, there were not enough CA anionic groups to be completely adsorbed on the surface of the LPI-stabilized oil droplets. Therefore, 70/30% (*w/w*) of CA/Pr-E was selected to prepare the Se-E. Meanwhile, the ζ-potential of the oil droplets showed an increase when the CA content was augmented from 0.1 to 0.4% (*w/v*), thus reaching a constant value around −54 mV. This implies that oil droplet surfaces were saturated completely by the CA molecules Burgos-Díaz et al. [14] points out that ionic biopolymer molecules can saturate the surfaces of lupin protein-coated oil droplets, generating stabilization of ζ-potential values. Once the addition of CA/Pr-E needed to produce Se-E was determined, the effect of adding CHI to form Te-E was tested (Figure 3b). In this regard, the ζ-potential value of the oil droplets did not show evidence of significant alterations (*p* > 0.05) when the addition of CHI in Se-E was increased from 50/50 to 80/20% (*w/w*), thereby attaining a constant value around 50.4 mV with 50/50% (*w/w*). This effect is like the adsorption of cationic CHI molecules on the surfaces of the Se-E oil droplets given the strong electrostatic force of attraction between them. As several authors have reported, colloidal systems presenting high ζ-potential values, greater than +25 mV or less than −25 mV, show, predominantly, great stability because of strong repulsive electrostatic forces between the droplets [33,34]. It would be expected, in this sense, to have high stability with respect to the aggregation due to the electrostatic repulsion derived from its high electric charge.

The physical stability of Pr-E, Se-E, and Te-E was examined based on the CI during storage for 7 days (Figure 4). 

The formation of cream is a result of differences in density between the dispersed phase and the dispersion medium. The CI (%) consists of the separation of two phases in the emulsion, characterized by the formation of a cream layer in the upper part of the sample tube and a clarification zone in the lower part of the tube [24]. The results revealed that the physical stability of Pr-E was similar to Se-E and Te-E until the first day of storage. Nonetheless, from day 1, a small rise in the CI of Pr-E was perceived compared to multilayer emulsions as a result of the movement of the oil drops by the action of the gravitational force between the lower and upper areas of the test tube. Pr-E indicated a process of destabilization through creaming that was demonstrated by a CI of 8.3 ± 2.1% after 7 days of storage, whereas the Se-E and Te-E emulsions showed a CI of 6.5 ± 1.0%. Therefore, emulsions stabilized with two and three interfacial layers of ionic biopolymers showed a slight improvement in physical stability compared to the emulsion stabilized with only one interfacial layer of LPI. According to Burgos-Díaz et al. [4], the higher multilayer emulsions stability stabilized with two and three interfacial layers might be mainly caused by a higher electrostatic repulsion between the droplets and by more efficient adsorption at the emulsion interface. Also, the deposition of several interfacial layers on the oil drops enhances the physical stability of the o/w emulsion by protecting it against cream formation due to repulsive forces between the oil drops and the increase in viscosity in the emulsion [33]. Thus, the results show that the stability of Pr-E is enhanced by Se-E and Te-E.

The optical microscopy images (Figure 5) revealed that the droplet size of the emulsions fluctuated slightly between multilayer emulsions (Se-E and Te-E) and the emulsion stabilized with only one layer (Pr-E).

In Figure 5a, it is shown that the droplet size of Pr-E was smaller than emulsions stabilized with two and three interfacial layers (droplet size mean value of Pr-E was 0.87 ± 0.07 µm). In addition, there are drops homogeneously distributed with very few aggregations between them, which indicates adequate physical stability of Pr-E. This could be ascribed to the high concentration of small-sized droplets in the Pr-E because of the high-pressure homogenization process and the presence of LPI as an emulsifier causing high electrostatic repulsion between the emulsion droplets. High electrostatic repulsion between LPI-stabilized oil droplets decreases the aggregations between them, thereby increasing the physical stability of the emulsion [24]. Furthermore, Figure 5b,c shows the droplets of the Se-E and Te-E, in which there is a slight increment in the size of particles and a wall thickness of the microcapsules compared to Pr-E given the presence of a second and third interfacial layer over the droplets of the emulsions with particle sizes of 7.22 ± 0.71 µm and 8.22 ± 2.65 μm (Zetasizer results), respectively. Similar behavior was observed by Burgos-Díaz et al. [14] in multilayer o/w emulsions since as the number of layers in the emulsion increased, the particle size also slightly increased. Emulsions that have small droplet sizes tend to be more stable during the spray-drying process, which is fundamental to achieve high retention of astaxanthin [35].

### 3.4. Effect of Environmental Stresses on the Emulsion Stability

Emulsions constitute systems that are thermodynamically unstable, which can be affected by environmental conditions commonly found during food processing like pH and salt concentration [14]. Consequently, the effects of pH and ionic strength (NaCl) on the emulsion stability was assessed (Figure 6). 

The experiment aimed to analyze the effect of pH on emulsion stability (Figure 6a). Changes in pH can strongly affect the stability of an emulsion, and it is essential to evaluate its effect on the formation of multilayer emulsions [9]. As expected, our results indicate that ζ-potential depended on the pH. 

Pr-E ζ-potential varied from 33.1 ± 3.89 mV to −32.2 ± 3.25 mV when the pH increased from 3 to 7, because of the amphoteric nature of proteins [8]. Moreover, at pH 5 in the Pr-E ζ-potential reached a change of 0 mV, which concurs with the isoelectric point (pH 4.65) of the LPI [14]; at this point, a phase separation was observed. Therefore, the Pr-E preserves its physical stability when the preparation and/or processing pH is lower or higher than pH 5. Thus, the function of multilayer emulsions will be to maintain the physical stability over the entire pH range evaluated.

The Se-E showed that the ζ-potential value changed from −50.9 ± 1.20 mV at pH 3 to −58.4 ± 1.70 mV at pH 7. The Se-E stabilized by CA and LPI showed negatively charged ζ-potential values over the entire range of pH values, which led to a high electrostatic repulsion among the oil droplets covered with CA/LPI layers, enhancing the physical stability (without phase separation) of the Se-E in comparison to a Pr-E. Similar behavior was observed by Gu et al. [36] when secondary emulsions were prepared using a layer of protein (lactoglobulin) as the emulsifier and another negative layer of carrageenan, obtaining high stability in the emulsion when it was subjected to a broad range of pH (3 to 8).

The Te-E presented ζ-potential values of positive charge from 51.5 ± 1.41 mV at pH 3 to 4.3 ± 1.20 mV at pH 7. However, a decrease in the ζ-potential was observed at pH values over 4.3, causing destabilization (emulsion aggregation) of the Te-E. This charge loss has been attributed previously to the proximity to the chitosan pKa (6.5) causing neutralization of NH_3_^+^ groups [8]. Such behavior has been described by Burgos-Díaz et al. [14], who evaluated the effect of a chitosan layer (secondary emulsion) on the surface of lupin protein-coated drops at a pH between 3 and 7. In conclusion, the Se-E was the most stable compared to the Pr-E and Tr-E when subjected to different pHs.

Figure 6b presents the effect of NaCl concentration on emulsion stability through changes in electrical charge (ζ-potential) on the surface of oil droplets. The presence of salts can alter the interfacial properties of emulsions as a result of a range of physicochemical mechanisms, along with the modification of the amount of polyelectrolyte absorbed, modifying the interfacial layer structure, or regulating the strength and extent of colloidal interactions between the droplets [9]. The results indicated that when there was an increase of NaCl concentration in all the emulsions studied, the electrical charge of droplets remained relatively constant without showing any destabilization in the emulsions. These results imply that the addition of salts did not alter the balance between the attractive (e.g., van der Waals, depletion, and hydrophobia) and repulsive (e.g., electrostatic, steric, and thermal fluctuation) interaction forces [9].

### 3.5. The Retention of Astaxanthin in the Emulsions during the Spray-Drying Process

Figure 7 shows the effect of the inlet temperature of the dried air (120 and 160 °C) on astaxanthin retention in emulsions during the process of spray-drying. 

The results showed that the retention of astaxanthin from Se-E and Te-E dried at 120 and 160 °C incremented considerably (*p* < 0.05) when compared to Pr-E, showing the protective effect of multilayer emulsions. Also, no relevant differences were observed (*p* > 0.05) in the retention of astaxanthin between Se-E and Te-E. In addition, after the spray-drying process at 120 and 160 °C, the powders obtained from the Pr-E, Se-E, and Te-E emulsions indicated retention rates of 88.5 and 87.4%, 96.4 and 95.3%, and 96.8 and 96.0%, respectively. The astaxanthin retention values of the multilayer emulsions were higher compared to those provided by Shen and Quek [35], who reported that the retention of astaxanthin in the o/w emulsions (milk proteins and soluble fibers of corn as encapsulation layers) was approximately 70% using an air inlet temperature of 160°C in the spray dryer. Fang et al. [34] reported an increase of β-carotene retention during spray drying when using two-layer emulsions (octenyl succinic anhydridestarch and chitosan) as a carotenoid encapsulation system. According to Bustamante et al. [6], the high astaxanthin retention could be attributed to the small emulsion droplet size achieved, which enhances the carotenoid retention during spray-drying.

Moreover, there were no significant differences (*p* > 0.05) observed with respect to the air inlet temperatures of the dryer. This could be ascribed to the short exposure time of the emulsions during this spray-drying process. This technique has been used extensively in the food industry due to its short drying time and is therefore highly suitable for the dry heating of labile compounds [5]. According to Bustos-Garza et al. [25], the microencapsulation of astaxanthin using a spray drying process delays the degradation of astaxanthin at storage due to its low aqueous activity.

Finally, the use of multilayer emulsions is preferable in spray drying processes with high temperatures, offering advantages such as greater stability of the emulsion and greater retention of the bioactive. Consequently, the use of this system could be an ideal option to produce dry emulsions, especially in the encapsulation of unstable bioactives such as astaxanthin.

## 4. Conclusions

This investigation revealed that stable multilayer o/w emulsions that contain astaxanthin microencapsulated can be produced using LPI, CA, and CHI as ionic interfacial layers. All emulsions stabilized with one, two, and three interfacial layers (Pr-E, Se-E, and Te-E) presented greater physical stability. However, the Se-E was the most stable compared to the Pr-E and Te-E when subjected to different pHs. Nevertheless, when the NaCl concentration was increased, no changes in the ζ-potential of any of the emulsions studied or destabilization were observed.

Se-E and Te-E exhibited higher astaxanthin retention during spray drying in comparison to Pr-E. Finally, it can be stated that the results obtained in this research have given us a broader understanding of multilayer emulsion systems in the retention of astaxanthin for its potential application in the food industry.

## Figures and Tables

**Figure 1 foods-10-00312-f001:**
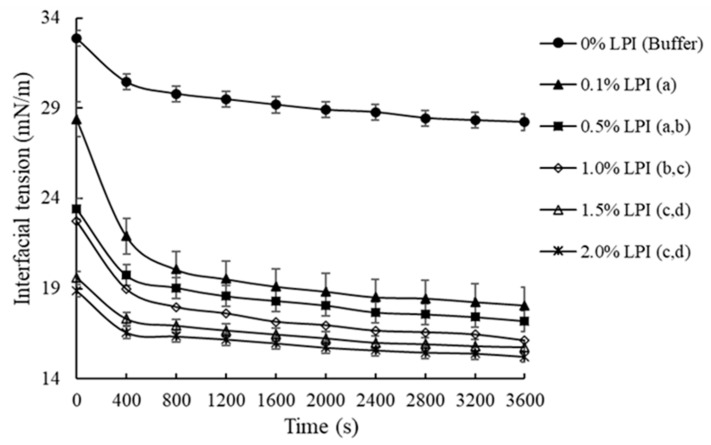
Interfacial Tension (IT) at oil/water interface containing different concentrations of lupin protein isolate (LPI). Different letters imply relevant differences (*p* < 0.05) in the LPI content.

**Figure 2 foods-10-00312-f002:**
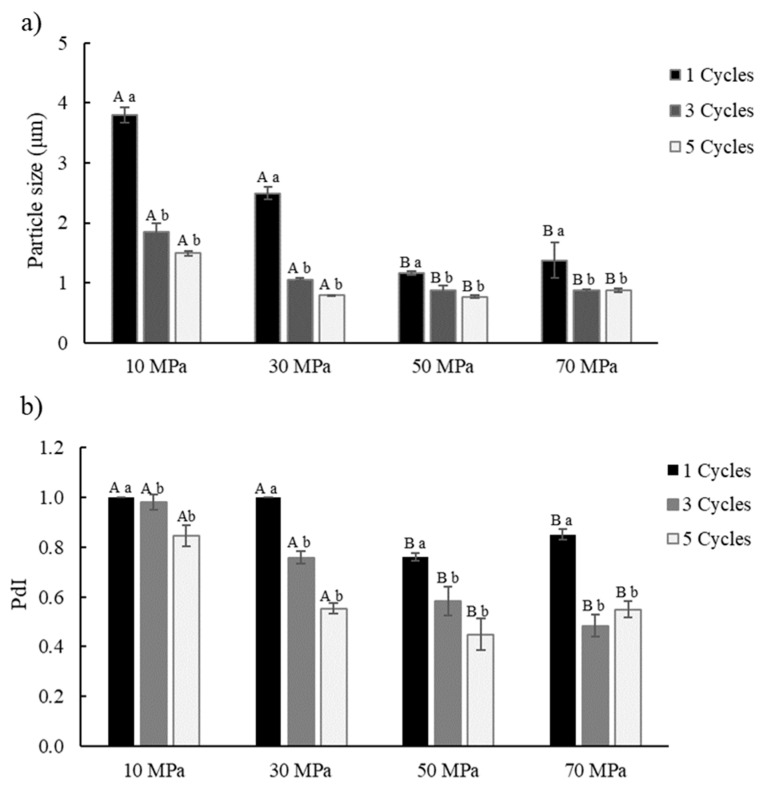
Effect of high-pressure homogenization (from 10 to 70 MPa) and the number of cycles (from 1 to 5) on the (**a**) particle size and (**b**) Polydispersity Index (PdI) of the primary emulsion. Different capital letters imply significant differences (*p* < 0.05) between homogenization pressures. Different lowercase letters show relevant differences (*p* < 0.05) between homogenization cycles.

**Figure 3 foods-10-00312-f003:**
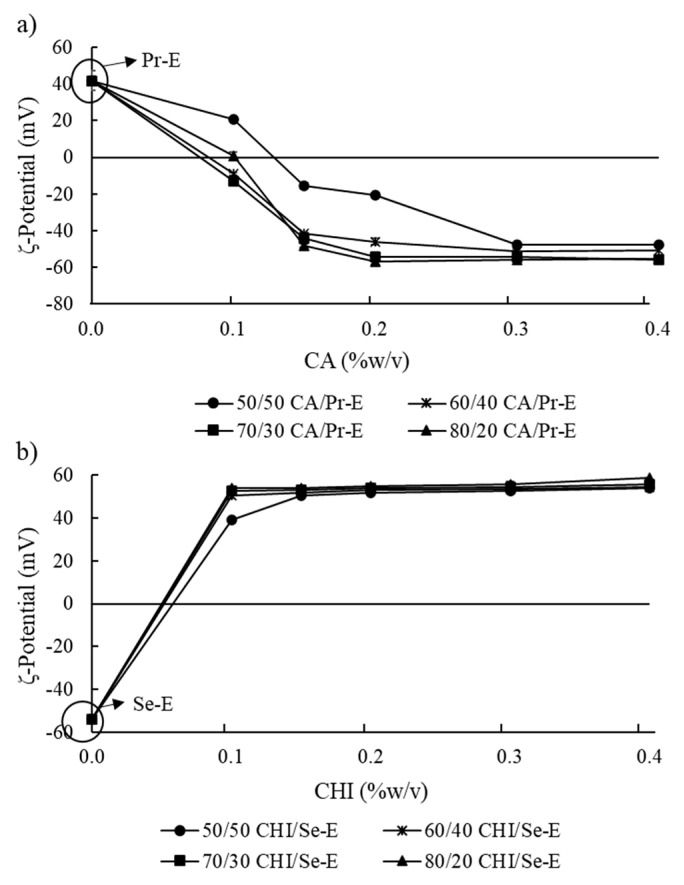
Effect of (**a**) ι-carrageenan (CA)/ Primary emulsion (Pr-E) and (**b**) chitosan (CHI)/secondary emulsion (Se-E) in different ratios (50/50 to 80/20% *w/w*) on the ζ-potential.

**Figure 4 foods-10-00312-f004:**
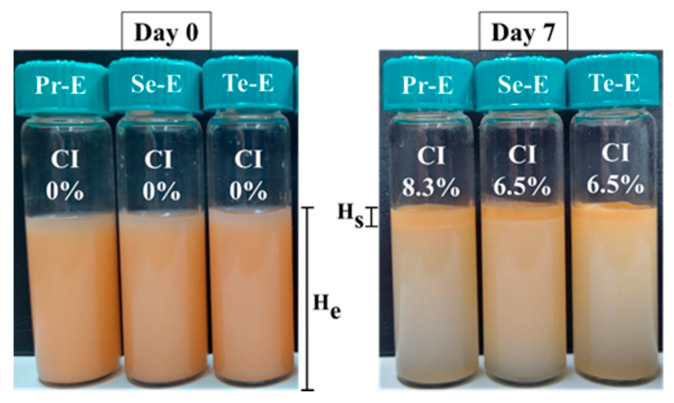
Evolution of Creaming Index (CI) (%) of Pr-E, Se-E, and tertiary emulsion (Te-E), during 7 days of storage at 25 °C.

**Figure 5 foods-10-00312-f005:**
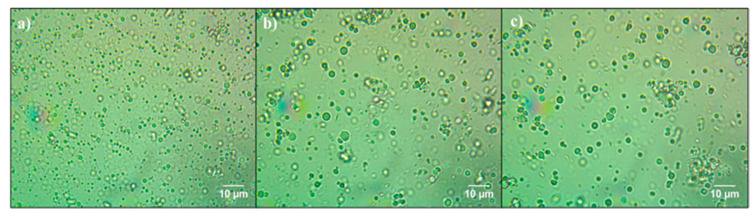
Optical microscopy picture at 100X of (**a**) Pr-E, (**b**) Se-E, and (**c**) Te-E.

**Figure 6 foods-10-00312-f006:**
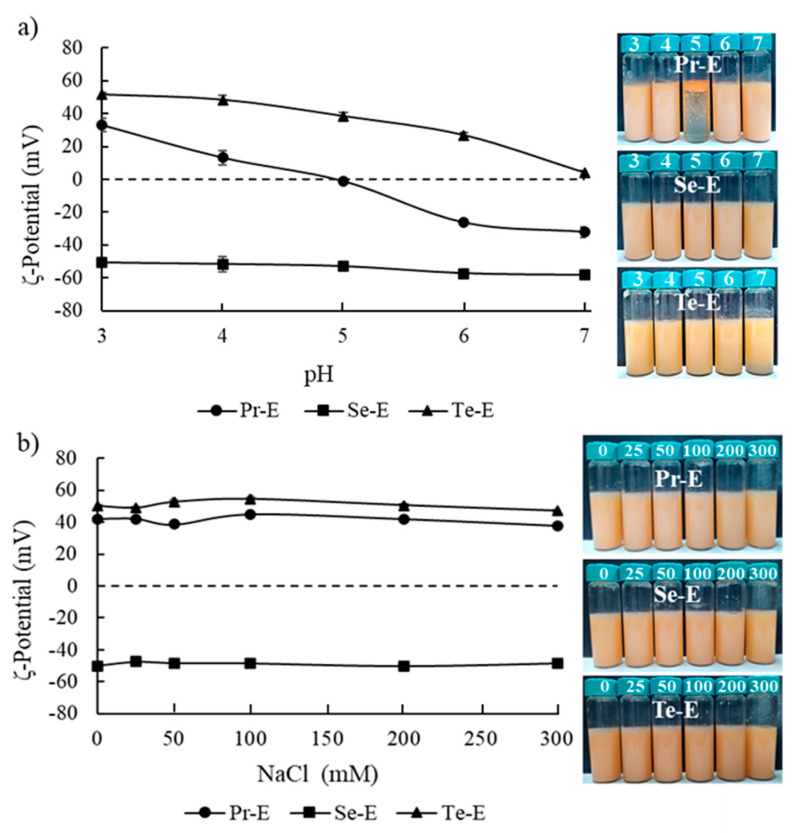
Effect of the (**a**) pH (3–7) and (**b**) NaCl (0–300 mM NaCl) on ζ-potential of emulsions.

**Figure 7 foods-10-00312-f007:**
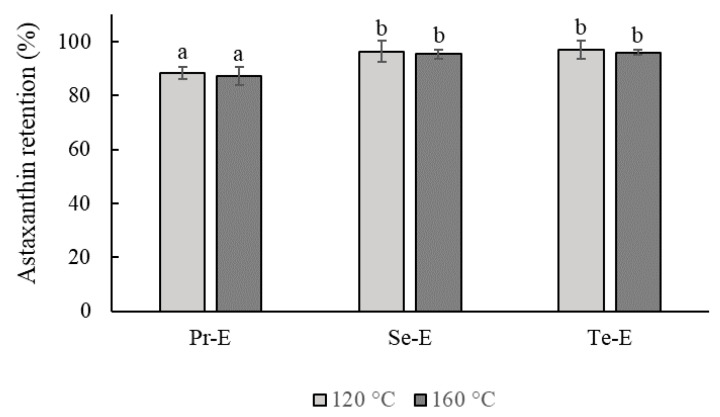
Effect of the inlet temperature of the dried air on the retention of astaxanthin of Pr-E, Se-E, and Te-E during the spray-drying process. Different letters represent significant differences (*p* < 0.05) in the retention of astaxanthin.

## Data Availability

Not applicable.

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
