# Peer review of "Effect of Interfacial Ionic Layers on the Food-Grade O/W Emulsion Physical Stability and Astaxanthin Retention during Spray-Drying"

_foods, 2021, doi:10.3390/foods10020312_

Round 1

Reviewer 1 Report

The manuscript is well-written and easy follow. The work fits the scope of the journal and there seem to be no major flaws in the experimental part.

Section 2.3 is not entirely clear to me; the authors describe that emulsions were made based on an established protocol, including some adjustments. Are the adjustments actually described in the experimental section of this work (they should be).

Also, the sentence starting at row 130 seems incorrect (style error).

Author Response

Reviewer 1:

Comments and Suggestions for Authors

The manuscript is well-written and easy follow. The work fits the scope of the journal and there seem to be no major flaws in the experimental part.

Section 2.3 is not entirely clear to me; the authors describe that emulsions were made based on an established protocol, including some adjustments. Are the adjustments actually described in the experimental section of this work (they should be).

Response: The section 2.3 was revised and modified in the manuscript.

Text modified:

2.3. Preparation of o/w multilayer emulsions with encapsulated astaxanthin

The oil phase of the emulsions was prepared with a 10% (w/w) astaxanthin and 90% (w/w) sunflower oil. The mixture was homogenized at 10,000 rpm during 5 minutes by means of a high speed blender (Benchtop homogenizer-ProScientific, USA), and afterwards it was passed through the high pressure homogenizer (HPH) (PandaPlus 2000, GEA Niro Soavi, Parma, Italy). The Pr-E was obtained by mixing a 10% (w/w) oil phase with 90% (w/w) protein emulsifier (from 0.1 to 2.0% w/v of LPI) in sodium citrate buffer at pH 3. The impact of HPH process conditions on the size of particles and the polydispersity index of Pr-E was determined. Different homogenization pressures (10 to 70 MPa) and the number of cycles (1 to 5) were analyzed.

The multilayer emulsions were prepared as follows: the secondary emulsion (Se-E) was prepared by mixing CA solutions (0.1 - 0.4% w/v, in sodium citrate buffer at pH 3) with Pr-E (CA/Pr-E) at different ratios (50/50, 60/40, 70/30, and 80/20% w/w). Then, tertiary emulsion (Te-E) was prepared by mixing CHI solutions (0.1 - 0.4% w/v, in sodium citrate buffer at pH 3) with Se-E (CHI/Se-E) at different ratios (50/50, 60/40, 70/30, and 80/20% w/w). The homogenization of Se-E and Te-E was carried out at 5,000 rpm during 2 minutes each one, followed by 2 cycles at 15 MPa through an HPH to cause the disruption of any flocs that could be formed. After that, the emulsions were homogenized at room temperature for 1 h by using a magnetic stirrer.

Also, the sentence starting at row 130 seems incorrect (style error).

Response: This information was revised and corrected in the manuscript.

Reviewer 2 Report

The manuscript presents an interesting analysis of  interfacial ionic layers effect on the emulsion physical stability and astaxanthin retention during the spray-drying process. The multilayer emulsions were stabilized  by ionic interfacial layers of lupin protein isolate (LPI), ι-carrageenan (CA), and chitosan (CHI).  Presented research results have a chance to find application field in industrial practice.

Writing style of whole article is clear and presented content is logically related. Graphics are easy to read and marked correctly. Bibliography is adequate to the topic, all included positions are cited in the text.

General Comments:

  1. In the introduction, the authors have comprehensively described the ionic layers used: carrageenan (CA) and chitosan (CHI). In contrast, astaxanthin is only briefly mentioned in lines 89-93. However, astaxanthin is a bioactive compound around which the whole idea of research and possible industrial application has arisen. Please extend your introduction to include more information on astaxanthin (natural sources, properties, use, etc.).
  2. Spray drying. Were there any special explanation for selecting maltodextrin as wall material ? What DE value of maltodextrin was used ?

Did the authors made a literature research on the efficacy of astaxanthin encapsulation by spray drying method for various wall materials ?

  1. Retention of astaxanthin in the emulsions during the spray drying process - results obtained by the authors should be compared more extensively with other researcher’s results. There are more articles in the literature on astaxanthin encapsulation by spray drying. The authors should provide all recent articles dealing with astaxanthin encapsulation and discuss the obtained results in the manuscript.

Sample articles on the topic:

Khalid, N., Barrow, C.J. (2018). Critical review of encapsulation methods for stabilization and delivery of astaxanthin. J. Food Bioact. 1: 104–115.

Bustos-Garza, C., Yáñez-Fernández, J., and Barragán-Huerta, B.E. (2013). Thermal and pH stability of spray-dried encapsulated astaxanthin oleoresin from Haematococcus pluvialis using several encapsulation wall materials. Food Res. Int. 54: 641–649.

Pu, J., Bankston, J.D., and Sathivel, S. (2011). Developing microencapsulated flaxseed oil containing shrimp (Litopenaeus setiferus) astaxanthin using a pilot scale spray dryer. Biosyst Eng. 108: 121–132.

Martínez-Álvarez, Ó., Calvo, M. M., & Gómez-Estaca, J. (2020). Recent Advances in Astaxanthin Micro/Nanoencapsulation to Improve Its Stability and Functionality as a Food Ingredient. Marine drugs, 18(8), 406.

  1. Please check carefully the test results for particle size and PdI, especially the statistical analyzes. Some results raise doubts, e.g. particle size 1 cycle 10 and 30 MPa or PdI 10 MPa 1 and 3 cycle.

Detailed comments:

Line 27: ‘… at 500 MPa …’ – it should be: ‘… at 50 MPa …’

Line 105: ‘… by CGNA.’ -  please explain the abbreviation in brackets.

Line 105: ‘… ι-carrageenan was obtained’ - what do '...' mean ?

Line 286: ‘… protein isolate [27] (Ma et al. , 2020) and lentil protein …’ - please correct citation in accordance with journal requirements.

Line 306: ‘Effect of CA/Pr-E and (b) CHI/Se-E …’ – it should be: ‘Effect of (a) CA/Pr-E and (b) CHI/Se-E …’  

Line 416:  ‘However, a decrease in the ζ-potential was observed at pH values over 5, …’ - the plot ζ-potential = f (pH) in Figure 6a shows that this decrease is already above pH 4, not pH 5.

Author Response

Reviewer 2:

Comments and Suggestions for Authors

The manuscript presents an interesting analysis of interfacial ionic layers effect on the emulsion physical stability and astaxanthin retention during the spray-drying process. The multilayer emulsions were stabilized by ionic interfacial layers of lupin protein isolate (LPI), ι-carrageenan (CA), and chitosan (CHI).  Presented research results have a chance to find application field in industrial practice.

Writing style of whole article is clear and presented content is logically related. Graphics are easy to read and marked correctly. Bibliography is adequate to the topic, all included positions are cited in the text.

General Comments:

  1. In the introduction, the authors have comprehensively described the ionic layers used: carrageenan (CA) and chitosan (CHI). In contrast, astaxanthin is only briefly mentioned in lines 89-93. However, astaxanthin is a bioactive compound around which the whole idea of research and possible industrial application has arisen. Please extend your introduction to include more information on astaxanthin (natural sources, properties, use, etc.).

Response: According to the reviewer’s suggestion, more information was incorporated in the manuscript.

Text included:

Astaxanthin is a xanthophyll carotenoid which is found in various microorganisms and marine animals. The natural sources of astaxanthin are microorganisms, phytoplankton, marine animals, and seafood such as shrimp, lobster, asteroidean, algae, fish, crustacean, trout, krill, red sea bream, and salmon [21]. Nowadays, studies on astaxanthin have increased due to the demand for natural bioactive compounds in the promotion of human health. The antioxidant activity of astaxanthin is 10 times more than zeaxanthin, lutein, canthaxanthin, β-carotene and 100 times higher than α-tocopherol. It protects organisms against a wide range of diseases with excellent safety and tolerability. Astaxanthin has great potential in food, nutraceutical and pharmaceutical applications [22]. Nevertheless, its applications in different food formulations are rather reduced since this carotenoid degrades rapidly during processing [24].

References:

  1. Fakhri S.; Abbaszadeh F.; Dargahi L.; Jorjani M. Astaxanthin: A mechanistic review on its biological activities and health benefits. Pharmacol. Res. 2018, 136, 1-20.
  2. Rao, R.; Moi, S.; Ravi, S.; Gokare, R. Astaxanthin: Sources, Extraction, Stability, Biological Activities and Its Commercial Applications—A Review. Mar. Drugs. 2014, 12, 128-152.
  3. Spray drying. Were there any special explanation for selecting maltodextrin as wall material? What DE value of maltodextrin was used?.

Response: According to the reviewer’s question, maltodextrin was used in this study due to its low viscosity to high concentrations (in water) and low cost. Normally, the incorporation of this polymer in the emulsion is carried out to increase the content of “total solids” of the sample and to increase the percentage of powder recovery (yield) during the spray drying process.  Therefore, this polysaccharide is widely in the food industry for performing the spray-drying.

 The Dextrose Equivalent (DE) of maltodextrin was between 16.5 and 19.5.  This information was incorporated in “Material and Methods” of this manuscript.

Did the authors made a literature research on the efficacy of astaxanthin encapsulation by spray drying method for various wall materials?

Response: In this study, the objective of this manuscript was to evaluate the effect of interfacial ionic polymer layers (lupin protein isolate, ι-carrageenan, and chitosan) on the emulsion physical stability and astaxanthin retention during the spray-drying process. Therefore, the encapsulation efficiency of astaxanthin with other wall materials was not evaluated since changes in this parameter (wall material) could interfere with the results of the influence of interfacial ionic polymer layers.

Retention of astaxanthin in the emulsions during the spray drying process - results obtained by the authors should be compared more extensively with other researcher’s results. There are more articles in the literature on astaxanthin encapsulation by spray drying. The authors should provide all recent articles dealing with astaxanthin encapsulation and discuss the obtained results in the manuscript.

 Sample articles on the topic:

 Khalid, N., Barrow, C.J. (2018). Critical review of encapsulation methods for stabilization and delivery of astaxanthin. J. Food Bioact. 1: 104–115.

Bustos-Garza, C., Yáñez-Fernández, J., and Barragán-Huerta, B.E. (2013). Thermal and pH stability of spray-dried encapsulated astaxanthin oleoresin from Haematococcus pluvialis using several encapsulation wall materials. Food Res. Int. 54: 641–649.

Pu, J., Bankston, J.D., and Sathivel, S. (2011). Developing microencapsulated flaxseed oil containing shrimp (Litopenaeus setiferus) astaxanthin using a pilot scale spray dryer. Biosyst Eng. 108: 121–132.

Martínez-Álvarez, Ó., Calvo, M. M., & Gómez-Estaca, J. (2020). Recent Advances in Astaxanthin Micro/Nanoencapsulation to Improve Its Stability and Functionality as a Food Ingredient. Marine drugs, 18(8), 406.

 Response: According to reviewer’s suggestion, two of reference were incorporate in the manuscript:

Text included in the Results and Discussion :

(1) According to Bustamante et al. [6], the high astaxanthin retention could be attributed to the small emulsion droplet size achieved, which enhances the carotenoid retention during spray-drying.

(2) According to Bustos-Garza et al. [26], the microencapsulation of astaxanthin by spray-drying retards its degradation during storage due to its low water activity.

  1. Please check carefully the test results for particle size and PdI, especially the statistical analyzes. Some results raise doubts, e.g. particle size 1 cycle 10 and 30 MPa or PdI 10 MPa 1 and 3 cycle.

 Detailed comments:

Line 27: ‘… at 500 MPa …’ – it should be: ‘… at 50 MPa …’

Response: This information was changed in the manuscript.

Line 105: ‘… by CGNA.’ -  please explain the abbreviation in brackets.

Response: This information was changed in the manuscript.

Line 105: ‘… ι-carrageenan was obtained’ - what do '...' mean?

Response: This information was corrected in the manuscript.

Line 286: ‘… protein isolate [27] (Ma et al., 2020) and lentil protein …’ - please correct citation in accordance with journal requirements.

Response: (Ma et al., 2020) was eliminated from the manuscript.

Line 306: ‘Effect of CA/Pr-E and (b) CHI/Se-E …’ – it should be: ‘Effect of (a) CA/Pr-E and (b) CHI/Se-E …’  

Response: This information was changed in the manuscript.

Line 416: ‘However, a decrease in the ζ-potential was observed at pH values over 5, …’ - the plot ζ-potential = f (pH) in Figure 6a shows that this decrease is already above pH 4, not pH 5.

Response: This information was corrected in the manuscript.

Reviewer 2:

Comments and Suggestions for Authors

The manuscript presents an interesting analysis of interfacial ionic layers effect on the emulsion physical stability and astaxanthin retention during the spray-drying process. The multilayer emulsions were stabilized by ionic interfacial layers of lupin protein isolate (LPI), ι-carrageenan (CA), and chitosan (CHI).  Presented research results have a chance to find application field in industrial practice.

Writing style of whole article is clear and presented content is logically related. Graphics are easy to read and marked correctly. Bibliography is adequate to the topic, all included positions are cited in the text.

General Comments:

  1. In the introduction, the authors have comprehensively described the ionic layers used: carrageenan (CA) and chitosan (CHI). In contrast, astaxanthin is only briefly mentioned in lines 89-93. However, astaxanthin is a bioactive compound around which the whole idea of research and possible industrial application has arisen. Please extend your introduction to include more information on astaxanthin (natural sources, properties, use, etc.).

Response: According to the reviewer’s suggestion, more information was incorporated in the manuscript.

Text included:

Astaxanthin is a xanthophyll carotenoid which is found in various microorganisms and marine animals. The natural sources of astaxanthin are microorganisms, phytoplankton, marine animals, and seafood such as shrimp, lobster, asteroidean, algae, fish, crustacean, trout, krill, red sea bream, and salmon [21]. Nowadays, studies on astaxanthin have increased due to the demand for natural bioactive compounds in the promotion of human health. The antioxidant activity of astaxanthin is 10 times more than zeaxanthin, lutein, canthaxanthin, β-carotene and 100 times higher than α-tocopherol. It protects organisms against a wide range of diseases with excellent safety and tolerability. Astaxanthin has great potential in food, nutraceutical and pharmaceutical applications [22]. Nevertheless, its applications in different food formulations are rather reduced since this carotenoid degrades rapidly during processing [24].

References:

  1. Fakhri S.; Abbaszadeh F.; Dargahi L.; Jorjani M. Astaxanthin: A mechanistic review on its biological activities and health benefits. Pharmacol. Res. 2018, 136, 1-20.
  2. Rao, R.; Moi, S.; Ravi, S.; Gokare, R. Astaxanthin: Sources, Extraction, Stability, Biological Activities and Its Commercial Applications—A Review. Mar. Drugs. 2014, 12, 128-152.
  3. Spray drying. Were there any special explanation for selecting maltodextrin as wall material? What DE value of maltodextrin was used?.

Response: According to the reviewer’s question, maltodextrin was used in this study due to its low viscosity to high concentrations (in water) and low cost. Normally, the incorporation of this polymer in the emulsion is carried out to increase the content of “total solids” of the sample and to increase the percentage of powder recovery (yield) during the spray drying process.  Therefore, this polysaccharide is widely in the food industry for performing the spray-drying.

 The Dextrose Equivalent (DE) of maltodextrin was between 16.5 and 19.5.  This information was incorporated in “Material and Methods” of this manuscript.

Did the authors made a literature research on the efficacy of astaxanthin encapsulation by spray drying method for various wall materials?

Response: In this study, the objective of this manuscript was to evaluate the effect of interfacial ionic polymer layers (lupin protein isolate, ι-carrageenan, and chitosan) on the emulsion physical stability and astaxanthin retention during the spray-drying process. Therefore, the encapsulation efficiency of astaxanthin with other wall materials was not evaluated since changes in this parameter (wall material) could interfere with the results of the influence of interfacial ionic polymer layers.

Retention of astaxanthin in the emulsions during the spray drying process - results obtained by the authors should be compared more extensively with other researcher’s results. There are more articles in the literature on astaxanthin encapsulation by spray drying. The authors should provide all recent articles dealing with astaxanthin encapsulation and discuss the obtained results in the manuscript.

 Sample articles on the topic:

 Khalid, N., Barrow, C.J. (2018). Critical review of encapsulation methods for stabilization and delivery of astaxanthin. J. Food Bioact. 1: 104–115.

Bustos-Garza, C., Yáñez-Fernández, J., and Barragán-Huerta, B.E. (2013). Thermal and pH stability of spray-dried encapsulated astaxanthin oleoresin from Haematococcus pluvialis using several encapsulation wall materials. Food Res. Int. 54: 641–649.

Pu, J., Bankston, J.D., and Sathivel, S. (2011). Developing microencapsulated flaxseed oil containing shrimp (Litopenaeus setiferus) astaxanthin using a pilot scale spray dryer. Biosyst Eng. 108: 121–132.

Martínez-Álvarez, Ó., Calvo, M. M., & Gómez-Estaca, J. (2020). Recent Advances in Astaxanthin Micro/Nanoencapsulation to Improve Its Stability and Functionality as a Food Ingredient. Marine drugs, 18(8), 406.

 Response: According to reviewer’s suggestion, two of reference were incorporate in the manuscript:

Text included in the Results and Discussion :

(1) According to Bustamante et al. [6], the high astaxanthin retention could be attributed to the small emulsion droplet size achieved, which enhances the carotenoid retention during spray-drying.

(2) According to Bustos-Garza et al. [26], the microencapsulation of astaxanthin by spray-drying retards its degradation during storage due to its low water activity.

  1. Please check carefully the test results for particle size and PdI, especially the statistical analyzes. Some results raise doubts, e.g. particle size 1 cycle 10 and 30 MPa or PdI 10 MPa 1 and 3 cycle.

 Detailed comments:

Line 27: ‘… at 500 MPa …’ – it should be: ‘… at 50 MPa …’

Response: This information was changed in the manuscript.

Line 105: ‘… by CGNA.’ -  please explain the abbreviation in brackets.

Response: This information was changed in the manuscript.

Line 105: ‘… ι-carrageenan was obtained’ - what do '...' mean?

Response: This information was corrected in the manuscript.

Line 286: ‘… protein isolate [27] (Ma et al., 2020) and lentil protein …’ - please correct citation in accordance with journal requirements.

Response: (Ma et al., 2020) was eliminated from the manuscript.

Line 306: ‘Effect of CA/Pr-E and (b) CHI/Se-E …’ – it should be: ‘Effect of (a) CA/Pr-E and (b) CHI/Se-E …’  

Response: This information was changed in the manuscript.

Line 416: ‘However, a decrease in the ζ-potential was observed at pH values over 5, …’ - the plot ζ-potential = f (pH) in Figure 6a shows that this decrease is already above pH 4, not pH 5.

Response: This information was corrected in the manuscript.
